# The Transorbital Approach: A Comprehensive Review of Targets, Surgical Techniques, and Multiportal Variants

**DOI:** 10.3390/jcm13092712

**Published:** 2024-05-05

**Authors:** Matteo De Simone, Cesare Zoia, Anis Choucha, Doo-Sik Kong, Lucio De Maria

**Affiliations:** 1Department of Medicine, Surgery and Dentistry “Scuola Medica Salernitana”, University of Salerno, Via S. Allende, 84081 Baronissi, Italy; 2UOC of Neurosurgery, Ospedale Moriggia Pelascini, Gravedona e Uniti, 22015 Gravedona, Italy; gioiaoffice@gmail.com; 3Department of Neurosurgery, Aix Marseille University, APHM, UH Timone, 13005 Marseille, France; anis.c13@gmail.com; 4Laboratory of Biomechanics and Application, UMRT24, Gustave Eiffel University, Aix Marseille University, 13005 Marseille, France; 5Department of Neurosurgery, Samsung Medical Center, Sungkyunkwan University, School of Medicine, Seoul, Republic of Korea; neurokong@gmail.com; 6Division of Neurosurgery, Department of Medical and Surgical Specialties, Radiological Sciences and Public Health, University of Brescia, Piazza Spedali Civili 1, 25123 Brescia, Italy; luciodemaria@gmail.com; 7Division of Neurosurgery, Department of Clinical Neurosciences, Geneva University Hospitals (HUG), Rue Gabrielle-Perret-Gentil 4, 1205 Geneva, Switzerland

**Keywords:** transorbital approach, TOA, skull base, minimally invasive, surgical techniques, TONES, SETOA, future prospective, endoscopic skull base, multiportal

## Abstract

The transorbital approach (TOA) is gaining popularity in skull base surgery scenarios. This approach represents a valuable surgical corridor to access various compartments and safely address several intracranial pathologies, both intradurally and extradurally, including tumors of the olfactory groove in the anterior cranial fossa (ACF), cavernous sinus in the middle cranial fossa (MCF), and the cerebellopontine angle in the posterior cranial fossa (PCF). The TOA exists in many variants, both from the point of view of invasiveness and from that of the entry point to the orbit, corresponding to the four orbital quadrants: the superior eyelid crease (SLC), the precaruncular (PC), the lateral retrocanthal (LRC), and the preseptal lower eyelid (PS). Moreover, multiportal variants, consisting of the combination of the transorbital approach with others, exist and are relevant to reach peculiar surgical territories. The significance of the TOA in neurosurgery, coupled with the dearth of thorough studies assessing its various applications and adaptations, underscores the necessity for this research. This extensive review delineates the multitude of target lesions reachable through the transorbital route, categorizing them based on surgical complexity. Furthermore, it provides an overview of the different transorbital variations, both standalone and in conjunction with other techniques. By offering a comprehensive understanding, this study aims to enhance awareness and knowledge regarding the current utility of the transorbital approach in neurosurgery. Additionally, it aims to steer future investigations toward deeper exploration, refinement, and exploration of additional perspectives concerning this surgical method.

## 1. Introduction

The search for increasingly less invasive surgery has led to the development of new approaches in neurosurgery as well. In particular, the skull base region is the one that most lends itself to the delineation of new, increasingly less demolitive approaches. However, beyond the clear enthusiasm for new approaches, these new techniques require rigorous study to ensure that widespread application is in the best interest of the patients [1]. 

Minimally invasive endoscopic approaches are on their way to consolidating themselves as the gold standard choice for a majority of anterior and middle skull base lesions [2,3]. Nevertheless, the success of surgery of any branch cannot be solely attributed to the technically correct conduct of the type of approach. A central role is played by the selection of the patient to whom you offer that approach.

Zada et al., in 2011, published their experience with EEA, attempting to define post hoc in which patients the endonasal approach has inherent limitations related to suboptimal patient selection. Among their findings, they had seen that significant suprasellar extension, lateral extension, retrosellar extension, cerebral invasion with edema, solid tumor consistency, involvement or vasospasm of the arteries of the circle of Willis, and engulfment of the optic apparatus or invasion of the optic foramina limited the extent of resection, with postoperative complications and the need for reintervention [4]. Therefore, preoperative evaluation is a key step in this process; indeed, in the aforementioned cases, those features could lend support to the choice of ab initio open craniotomy.

Initially limited to the treatment of orbit-related pathologies, transorbital approaches (TOAs) have expanded their utility to encompass a broader spectrum of lesions within the skull base, independently or in combination with trans-nasal techniques. This evolution has allowed surgeons to resect various pathologies while avoiding the more extensive and potentially disfiguring trans-facial or transcranial methods. The term “endoscopic transorbital approach” (ETOA) encapsulates a number of endoscopic surgical routes suitable for addressing various lesions in the anterior and middle cranial fossae. The TOA provides access to various regions of the orbit and skull base. This approach involves navigation through the orbit, a bony cavity that houses the eye and associated structures, including extraocular muscles, nerves, and blood vessels. Within the skull base, the TOA allows for the exploration of the middle cranial fossa, which houses structures such as the temporal lobes and Meckel’s cavern, and the anterior cranial fossa, which houses critical elements such as the optic nerves and olfactory bulbs. In addition, the TOA facilitates access to the petrous apex, a pyramid-shaped bony prominence that houses important neurovascular structures. This approach also provides a pathway to the cavernous sinus, a venous sinus located on each side of the sella turcica, which serves as a conduit for crucial nerves and blood vessels. Understanding the anatomical landmarks and relationships in these regions is essential for the safe and effective use of the transorbital approach in surgical interventions.

Mastery of the ETOA requires a multifaceted understanding of surgical anatomy, as it traverses regions typically accessed by different routes by different specialists, requiring the identification of anatomical landmarks from an endoscopic perspective. The advantages of mastering this technique include reduced morbidity, no visible scarring or external craniotomies, and minimal brain manipulation. Good preoperative selection also comes through a thorough study of the anatomy; an EANS survey on experienced skull base surgeons showed that most respondents use both MRI and CT scans preoperatively to study the anatomy [5,6]. 

Recognizing and defining surgical goals before intervention is crucial, considering factors like underlying pathology, patient age, and general health [7]. Tumor anatomy evaluation is fundamental, and total resection may not always be feasible due to the proximity to vital structures or the outweighing of surgical risks over benefits [8]. While endoscopic approaches are expanding, they may not be suitable for pathology close to critical neurologic and vascular structures [9]. The biological nature of the tumor, especially in malignant cases, should be considered in surgical planning. The impact on quality of life is significant, and joint decision-making between physicians and patients is integral in determining surgical goals [10]. Overall, the prospects of using these approaches in combination with each other and the considerable variations developed on the individual approaches, as well as the addition of complementary techniques such as radiosurgery [11], offer a considerable range of possibilities that move in the direction of personalizing treatment options. Based on these assumptions, it is essential to first clarify the pathologies that can be easily treated by the approach, presenting them in a framework of surgical strategies that may also include the combination of multiple minimally invasive approaches presented in ascending order of complexity. The purpose of this work is to examine the efficacy and safety of the TOA in treating a wide range of neurosurgical pathologies, focusing on different surgical targets and their respective complications, as well as comparing outcomes with other traditional surgical approaches. Additionally, the aim is to assess the complexity of TOA procedures and provide an overview of various techniques and tools used to address specific pathologies in different anatomical segments.

## 2. Transorbital Route, Surgical Targets, and Corridors

The transorbital approach (TOA) represents a valid minimally invasive alternative to a multitude of surgical routes and has the potential to safely address a wide range of pathologies [12]. The TOA has been studied for multiple targets, each requiring a different set of neurosurgical skills and different levels of complexity. According to a proposed scale of difficulty by Di Somma et al., there are five stages in transorbital surgeries: the extraconal and extradural corridor (stage 1): the intradural one (stage 2): the intraconal, Meckel’s cave interdural, and anterior temporal lobe approaches (stage 3): the opticocarotid, cavernous sinus, and mesial temporal region route (stage 4): and finally the petrous apex, posterior cranial fossa, insula, and Sylvian fissure approaches (stage 5) [13]. In our work, we review the aforementioned targets through the transorbital approach, adhering to the proposed classification of complexity.

### 2.1. Extraconal Approach

The transorbital approach facilitates the resection of lesions within the extraconal space, defined as the area inside the orbit but outside the musculofascial cone bordered by the four recti muscles. Target lesions in this region include extra-ocular muscle herniations and metastases. Additionally, the transorbital approach enables access to extradural lesions of the middle and anterior cranial fossa, such as spheno-orbital meningiomas and other extradural tumors, as well as surgical interventions for fractures with cerebrospinal fluid (CSF) leak. An anterior orbitotomy is typically favored as the entry point for transorbital resection of extraconal lesions, with the combination of anterior orbitotomy and superior osteotomy being most suitable for larger lesions [14].

### 2.2. Extradural Approach for Middle or Anterior Fossa

A growing body of literature supports the effectiveness and safety of the transorbital approach for treating spheno-orbital (SO) meningiomas. One review, analyzing 65 cases, noted significant improvements in various clinical aspects such as reduced proptosis (observed in all cases), improved visual deficits, and ocular paresis. Similarly, another review of 58 patients reported comparable positive outcomes, including reductions in proptosis and visual impairment. Complications, though relatively rare, included trigeminal dysesthesia, CSF leak, and transient ophthalmoplegia. Overall, the transorbital approach proves particularly beneficial for SO meningiomas, especially when addressing specific tumor characteristics or as part of a multistage treatment plan [15]. The TOA can be safely applied to resect meningiomas originating in the olfactory groove. This surgical approach, indeed, allows for minimally invasive access, avoidance of brain retraction, and ease for early tumor devascularization. A case reported in the literature described a gross total resection of a large WHO I olfactory groove meningioma, through the left-sided transorbital approach, allowing for a gross total resection without surgical complications [16].

A systematic review and meta-analysis involving 59 studies and 1903 patients compared the transorbital approach to the microsurgical transcranial approach for spheno-orbital meningioma (SOM) resection. The findings revealed a lower rate of gross total resection with the transorbital approach. However, this approach exhibited fewer cranial nerve focal deficits and yielded better outcomes in reducing proptosis and improving visual acuity. Notably, there were no significant differences in progression-free survival rates between the two surgical approaches [17].

Although less common than intraconal pathologies, cavernous venous malformations (CVMs) within the extraconal space can be addressed and treated via the transorbital route. A reported case in the literature described the successful treatment of an extraconal CVM using this approach. The noted advantages included excellent exposure of both extraconal and intraconal compartments, superior illumination and visualization, a direct and efficient route to the target, and minimized manipulation of bone and muscle, thereby reducing damage to normal structures [18,19].

### 2.3. Intraconal 

The transorbital approach can be used to reach intraconal lesions of various kinds. From a technical point of view, it was suggested that the most convenient entry point to reach intraconal lesions, via the transorbital approach, is the medial orbitotomy for well-circumscribed intraconal lesions and the combined medial–lateral transorbital approach for larger lesions [14]. An example of intraconal lesion potentially resectable via transorbital is the intraconal meningioma of the orbital apex. The reported benefits, from a review collecting 24 cases operated thought the lateral trans-eyebrow endoscopic TOA (ETOA), were increased illumination and magnification of the surgical field and an unparalleled lateral view of the orbital apex [20]. Regarding intraconal pathology, however, it must be remembered that lesions in the inferomedial quadrants are approached with the EEA, while those in the superolateral and inferolateral quadrants are approached with the ETOA. For the approach to superomedial lesions, however, the EEA can be used in combination with the ETOA [21]. 

### 2.4. Meckel’s Cave

The TOA has been utilized to access lesions within Meckel’s Cave, facilitated by cadaveric anatomical studies [22,23]. Additionally, a case report documented nine patients with lesions in Meckel’s Cave and the middle cranial fossa, comprising four trigeminal schwannomas, two meningiomas, one metastatic brain tumor, one chondrosarcoma, and one dermoid cyst. Among these patients, seven (77.8%) underwent only the ETOA, while the remaining two underwent a combined ETOA and endoscopic endonasal approach or retrosigmoid craniotomy. GTR was achieved in seven out of nine patients. Regarding postoperative complications, only one patient experienced ptosis, which resolved after six months, and no cases of postoperative cerebrospinal fluid (CSF) leak were reported [24]. The utilization of the TOA in reaching lesions within Meckel’s Cave, as supported by anatomical studies and clinical cases, demonstrates its versatility and efficacy in addressing various pathologies in this complex anatomical region. The high success rate in achieving GTR and the low incidence of postoperative complications, particularly CSF leak, underscore the feasibility and safety of this approach.

### 2.5. Cavernous Sinus

The TOA is frequently used in treating carotid cavernous fistulas (CCFs). This method has shown effectiveness, particularly for dural carotid fistulas. A systematic review of 30 studies reported a high success rate (89.9%) for this procedure, with significant improvements in visual acuity (93.4%) and proptosis reduction (88.1%). Complications were minor and manageable, including subconjunctival hemorrhage, infraorbital hemorrhage, eyelid hematoma, and foreign-body granuloma [25]. These findings underscore the TOA’s efficacy and safety in managing complex pathologies within the cavernous sinus. A study assessing the benefits of venous transorbital access via the left angular vein, distal superior ophthalmic vein (SOV), and cavernous sinus revealed advantages such as avoiding damage to the internal carotid artery, distal migration of detachable balloons or thrombi, and intracavernous pseudoaneurysm formation. Notably, major complications were absent, and minor complications included subconjunctival hemorrhage, infraorbital hemorrhage, eyelid hematoma, and foreign-body granuloma. These findings highlight the efficacy and safety of the transorbital approach in managing CCFs, particularly through venous access routes [26]. 

### 2.6. Mesial Temporal Lobe Epilepsy 

The transorbital approach has been described from anatomical studies on four cadavers (eight orbits), to the successful exposure of the medical temporal artery and mesial temporal lobe [27]. This made this approach a valid candidate to address pathologies of these territories, such as tumors or tissue resection in the context of mesial temporal lobe epilepsy. Mesial temporal lobe epilepsy is an important cause of drug-resistant epilepsy, for which surgery is the curative option, traditionally involving a frontotemporal craniotomy and open resection of the anterior temporal cortex and mesial temporal structures. The transorbital approach has been investigated among the potentially minimally invasive approaches, along with endoscopic trans-maxillary, endoscopic endonasal, endoscopic transorbital, and endoscopic supracerebellar trans-tentorial, to highlight the potential of these endoscopic techniques compared to the traditional surgeries [28]. However, we recall that, in this context, there are functional neurosurgical techniques that simultaneously meet the goals of mini-invasiveness and safety and efficacy, such as laser interstitial thermal therapy (LITT) [29].

### 2.7. Petrous Apex and Posterior Cranial Fossa Approach

In 2018, De Somma et al. described a surgical way of accessing the petrous apex bone through a transorbital endoscopic approach, setting a significant leap in minimally invasive procedures [30]. Their work detailed an innovative dissection method supported by 3D imagery and quantitative bone removal analysis, successfully navigating beyond previous boundaries. This four-handed technique starts with an upper eyelid crease incision, followed by an interdural dissection between the periorbita and temporal pole. Once the temporal lobe is elevated in extradural fashion and the middle meningeal artery cut, the greater superficial petrosal nerve is unveiled, acting as a landmark of the underlying petrous internal carotid artery. The temporal lobe is finally elevated and the trigeminal porus opened to further expose the petrous apex. 

Later that same year, Noiphithak et al. proposed two variations of this approach, namely the lateral transorbital approach (LTOA) and the lateral orbital wall approach (LOWA), and confronted them to the traditional transcranial anterior trans-petrosal approach (ATPA) in a cadaveric study with respective measurements of the area of exposure, surgical freedom, and angles of attack [31]. They found that the TOA using the lateral orbital corridor for posterior fossa (PF) access is a technique that may provide a comparable surgical exposure to the ATPA. Furthermore, the removal of the orbital rim showed an additional benefit in an enhancement of the surgical maneuverability in the PF.

In 2020, Topczewski et al. conducted a comparative anatomical study between the ventral EEA and the TOA for accessing the petrous apex [32]. They found that the TOA allowed for a 48.3% removal of the petrous bone at its most superolateral part, whereas the EEA facilitated a 48.7% removal at the inferomedial area. Both methods offered distinct visualization and access to the petrous apex and adjacent neurovascular structures. Significantly, they discovered a connection area between the two surgical paths, bordered by critical anatomical features such as the internal carotid artery and the abducens nerve. Utilizing these approaches in tandem, the researchers achieved a combined petrous apex removal efficiency of 97%. This multiportal strategy could be especially beneficial for treating lesions in the petrous apex and petroclival regions, particularly when traditional transcranial or sole endonasal endoscopic methods fall short.

In 2022, Lee et al. confronted the EEA and TOA in the clinical setting through a retrospective cohort of 19 patients operated for petrous lesions [33]. They categorized the petrous apex (PA) into three zones relative to the petrous segment of the internal carotid artery (p-ICA): above (zone 1), behind (zone 2), and below (zone 3). Their findings suggest that the EEA is effective for lesions across all PA zones, particularly for clival tumors extending medially to laterally into the PA. Meanwhile, the TOA offers direct access to PA’s superior region (zone 1) and may be preferable for cystic diseases or selected PA pathologies [30].

### 2.8. Approach to Insula and to Sylvian Fissure

The insula is an anatomically and functionally complex region, both because due to its deep location in the sylvian fissure, and due to the incompletely clarified functions entailed [34]. The resection of tumors located in the insula poses major concerns about the choice of the best surgical strategy, especially considering the proximity to the middle cerebral artery (MCA) during the procedure. Traditionally, these tumors were resected through the trans-sylvian or the transcortical routes. However, the transorbital approach was studied in four cadavers and, after being validated, it was successfully reproduced in a patient suffering from an insular glioma. Despite the limited evidence, this approach was presented as a new surgical corridor to access the insular region, especially beneficial for lesions in the anterior part of the insula. In contrast to other surgical corridors, the transorbital dissection, in this case, allows for subpial tumor dissection with minimal manipulation of the MCA and its branches. A disadvantage of the TOA is that, along with the minimal invasiveness, there is limited space available for the maneuverability of surgical instruments and therefore the ability to control bleeding complications [35].

## 3. Surgical Techniques

### 3.1. Open Microsurgical 

The integration of advanced instruments into micro-neurosurgery has improved the safety of various neurosurgical procedures, reducing the risk of severe complications [36]. In a recent study by Houlihan et al [37]., the focus was on evaluating the practicality and effectiveness of a biportal bitransorbital approach. This study, conducted on ten cadaver specimens, compared three different approaches: midline anterior subfrontal (ASub), bilateral transorbital microsurgery (bTMS), and bilateral ETOA (bETOA). The measurements included the lengths of cranial nerves I and II, the optic tract, A1, and the exposure area of the anterior cranial fossa floor. The study also analyzed angles of attack (AOAs) and volume of surgical freedom (VSF) to determine instrument maneuverability. The goal was to assess whether the biportal approach offered greater freedom, especially around critical structures like the bilateral paraclinoid internal carotid arteries (ICAs), bilateral terminal ICAs, and anterior communicating artery (ACoA). Challenges were encountered with both bTMS and bETOA in reaching bilateral A1 segments and the ACoA. The study found comparable total frontal lobe exposure areas (AOEs) between ASub (1648.4 mm²), bTMS (1658.9 mm²), and bETOA (1914.9 mm²), with no significant difference observed (*p* = 0.28). Overall, both microscopic and endoscopic approaches demonstrated advantages. For terminal ICA access, bTMS and ASub yielded similar results, and all three approaches (bTMS, bETOA, and ASub) showed equivalent maneuverability. This study, alongside others by Houlihan et al. (2022), supports the effective use of a transorbital approach to reach the terminal ICA. The maneuverability of instruments in the transorbital microsurgery corridor consistently impresses, suggesting its consideration in terminal ICA lesion surgeries. While concerns about crowding in the endoscopic corridor led to the proposal of a biportal approach, a detailed comparison revealed that the microscopic technique within the same transorbital corridor offers superior surgical freedom, regardless of biportal visualization and access.

### 3.2. Endoscopic

Endoscopic procedures in the orbital and intracranial regions involve the use of 4 mm endoscopes with varying angles, including 0°, 30°, 45°, 70°, and occasionally 120°. The 0° endoscope is the primary choice for the majority of the dissection [38]. The development of endoscopic surgery brought an improved magnification, illumination, and visualization of the surgical field [39]. The use of the endoscopic transorbital approach evolved to minimize invasiveness and surgical complications in open skull base surgery, while maintaining the adequate standards of visibility [40]. The ETOA is a type of endoscopic surgery using orbitotomies for various pathologies and indications. It is a true TOA that does not involve removing the orbital rim or frontal bone. Proposed as a superior way to access the far lateral anterior and middle skull base compared to trans-nasal approaches [41], this method is attractive because it comes with minimal complications, leaves no visible scars, requires a small craniotomy, and involves limited brain retraction. This approach minimizes collateral damage to nearby structures, promoting quick patient recovery, potentially avoiding extended ICU stays, and reducing the need for prolonged pain medication [38]. In their comprehensive systematic review, Vural et al. [22] reported that the most common neurosurgical pathologies addressed via ETOA approaches include meningiomas (45% of cases), CSF leaks (15.4%), inflammatory or infectious processes including abscesses (11.4%) and schwannomas (6.7%). Moreover, among the 102 tumor cases analyzed in their study, the use of TONES techniques resulted in a 49% rate of gross-total resection, 8.8% near-total, 29.4% subtotal, and 5.9% partial resection, with only 19 cases undergoing adjuvant radiation treatment. The endoscopic approach aims to create a coplanar endoscopic surgical channel for better vision and magnification of the pathology. The surgical channel passes through a craniotomy made through one of the four orbital walls and progresses into the orbit. This approach introduced the concept of the sino–orbito–cranial interface, a crucial and surgically complex region [41,42]. There are four fundamental ETOA approaches, which correspond to the four orbital quadrants: the superior eyelid crease (SLC), precaruncular (PC), the lateral retrocanthal (LRC), and the preseptal lower eyelid (PS). (Table 1) These approaches make it possible to reach the surgical target without disrupting significant structures or causing functional impairment [38].

#### 3.2.1. Superior Eyelid Crease Approach

The SLC approach, also known as the upper eyelid approach, is the most common approach for the ETOA. With this method, it is possible to reach the superior orbit, frontal sinus, anterior skull base (ASB), supraorbital and posterior–central regions of the anterior cranial fossa (ACF), and the lateral regions of the middle cranial fossa (MCF) [22]. The skin is incised on the supratarsal fold, but it can be adjusted depending on the target [12,38], and eyebrow incision has also been described [43]. The dissection involves lifting the inner layer of the orbicularis oculi muscle toward the upper orbital rim to prevent the opening of the orbital septum and periorbital tissue, which could lead to fat protrusion into the surgical area. Once the orbital rim is identified, the periosteum is cut, and the dissection continues in a subperiosteal plane of the orbital roof. (Figure 1) After locating the optic canal on the posterior surface of the orbit and the ethmoid foramina medially, the dissection proceeds laterally. It is crucial to locate the superior orbital fissure (SOF) to obtain a proper orientation in the corridor’s most posterior and lateral aspects. To create a more lateral corridor, the lateral canthal ligament can be detached if necessary [12]. Once the location of the craniectomy is decided depending on the target location, a Diamond burr or a chisel is used to open the bone. Depending on the nature of the disease, the dura of the ACF and MCF can be elevated or incised [44,45].

Orbital rim is identified, the periosteum is cut, and the dissection continues in a subperiosteal plane of the orbital roof.

#### 3.2.2. Precaruncular Approach

With the PC approach, it is possible to achieve a direct and avascular approach to the medial orbital roof, lamina papyracea, cavernous sinus, ethmoidal arteries, parasellar and paraclinoid aspects of the internal carotid artery, along with the optic nerve and the anterior skull base [46,47,48,49]. An incision is made by cutting through the conjunctiva at the apex of the medial canthus, between the caruncle and the skin. Entry into the avascular plane occurs beneath Horner’s muscle and the posterior limb of the medial canthal tendon. The following incision is made in the periorbita at the level of the crista lacrimalis. The dissection then progresses from the anterior to the posterior, running between the periorbita and the medial orbital wall. The approximate level of the ASB can be assessed by identifying the ethmoidal bundles coursing along the frontoethmoidal suture, which can be cauterized and cut. The presence of the posterior ethmoidal artery indicates proximity to the optic nerve, so close attention must be paid during the surgical dissection. The dissection continues across the medial orbital wall until the orbital apex. At that point, the craniectomy is performed according to the location of the surgical target [12,22,38].

#### 3.2.3. Lateral Retrocanthal Approach

The LRC approach permits access to the deep lateral orbit, lateral aspect of the ACF, MCF, infratemporal, and temporal fossa [38,50]. An incision is made through the conjunctiva, posteriorly to the lateral canthus insertion. This method helps to prevent eyelid scarring and disruption [22]. Subperiosteal dissection is carried out along the lateral orbital wall, extending from the inferior orbital fissure (IOF) to the orbital apex, allowing for a better exposure of the greater sphenoidal wing (GSW) between the SOF and IOF, posterior to the zygomatic bone [22]. Removing the GSW allows for access to the MCF, temporal, and infratemporal fossae [38]. This approach bears no risk for optic nerve damage, as the optic nerve is separated from the surgical corridor by all SOF contents and the optic strut [22,38]. Alternative access, specifically to the lateral part of the frontal fossa, can be achieved through a craniectomy centered on the sphenofrontal suture, located in the superior aspect of the lateral orbital wall [22,38]. The LRC approach offers a solution to issues such as scarring and eyelid support disruption, commonly associated with cutaneous or canthotomy incisions [22,38]. Chibbaro et al. [51] demonstrated three variations of the lateral retrocanthal approach: the superomedial, superolateral, and inferolateral. The first enabled access to the optico-carotid area, cerebral falx, and medial rim of the sphenofrontal suture. Conversely, the superolateral and inferolateral approaches allowed for the exposure of the lateral anterior and middle cranial fossae, extending the surgical access to the Sylvian fissure posteriorly.

#### 3.2.4. Preseptal Lower Eyelid Approach

The PS approach serves as a valuable means to reach the inferior orbit. Its combination with LRC or PC further enhances maneuverability and optimizes exposure, respectively, in the lateral and medial orbital quadrants [38,45]. This approach provides a pathway through the orbital floor, the maxillary sinus, IOF, and the foramen rotundum. For a PS approach, the conjunctival incision is performed 2 mm below the tarsus and around 6 mm inferior to the eyelid margin, on the conjunctival surface of the lower eyelid. Once the orbicularis oculi is identified, the dissection proceeds along its posterior surface, which lies anteriorly to the inferior orbital septum. Following the orbicularis oculi muscle along the inferior orbital rim, the periosteum is incised and elevated off the orbital floor. Further dissection can be carried out by cutting the infraorbital bundle and IOF [22,38].

**Table 1 jcm-13-02712-t001:** Exposed areas via the four fundamental ETOA approaches.

Superior Eyelid	Precaruncular	Lateral Retrocanthal	Preseptal Lower Eyelid
EXPOSED AREAS			
- superior orbit - frontal sinus - anterior skull base (supraorbital and posterior part) - middle skull base	- medial orbital roof - lamina papyracea - cavernous sinus - ethmoidal arteries - parasellar and paraclinoid ICA - optic nerve - anterior skull base	- lateral orbit - infratemporal fossa - temporal fossa - lateral part of anterior cranial fossa - lateral part of middle cranial fossa - optico-carotid area (superomedial variation) - falx cerebri (superolateral variation) - sphenofrontal suture (inferolateral variation)	- inferior orbit - lateral and medial orbital quadrants (when combined with LRC or PC)

### 3.3. Complications and Reconstruction Methods

In their systematic review analyzing TOA techniques, Vural and colleagues highlight the fact that bony reconstruction is typically applied in case of large bony defects, and it is most commonly applied in multi-layer fashion [22]. Moreover, reconstruction has to be watertight in case of egress in structures beyond the orbital rim; for example, the sphenoid sinus, cribriform plate, planum sphenoidale, and lateral sphenoid recess [22,48].

Vural et al. [22] performed a comprehensive assessment of postoperative complications associated with TOA procedures. The analysis of 193 procedures highlighted 60 cases of postoperative complications, all of which were classified as Grade 1 or Grade 2, except one case of Grade 3b complication consistent with a periorbital pseudomeningocele, requiring surgical correction through shunting. Moreover, most of the reported complications were transient in the postoperative period. 

### 3.4. Exoscope

When working within confined anatomical spaces like the orbit, which houses numerous delicate structures, any procedures aimed at removing intraconic lesions—whether through trans-cranial or transpalpebral approaches—demand a magnified view of anatomical details. Traditionally, loupes or microscopes have been employed to fulfill this requirement. However, recently, the use of the exoscope has become more popular as an alternative or in support to the traditional microscope [52]. The use of an exoscope can bring significant advantages by increasing magnification potential by means of an integration of an optical and digital zoom, improved ergonomics and improved visualization angles of the operative anatomy [53]. 

A paper by Iwami et al. [54] presents two cases wherein a combination of endoscopic and exoscopic techniques was utilized to address medial temporal lobe (MTL) lesions through a lateral orbital wall approach (LOWA). In both cases, a three-dimensional exoscope provided observation of superficial areas and facilitated procedures requiring stereoscopic vision, while the endoscope aided in observing deeper areas and navigating narrow cavities where its use was more advantageous. The first case involved a left MTL glioblastoma, where an endoscope was initially employed for corticotomy and tumor removal but was later switched to a smaller endoscope for improved visibility. In the second case, another glioblastoma patient underwent dissection with an exoscope on the tumor’s anterior surface, while an endoscope was utilized for two-handed procedures to minimize damage and bleeding risk. Peron et al. [52] report on the use of a High-Definition 4K-3D exoscope in removing an orbital cavernous hemangioma through a transpalpebral approach. The innovative device provides high-resolution 4K-3D images, enhancing surgical precision with improved depth perception, color representation, and magnification. Despite the need for the surgeon and assistant to share the same monitor, the exoscope offers ergonomic advantages and seamless integration with other surgical techniques, making it a valuable tool in orbital surgery. The integration of navigation and ultrasound with the exoscope, facilitated by the ample space above the operating field, enhances the minimally invasive nature of the trans-eyelid approach by minimizing the manipulation of orbital intraconic structures. This not only yields optimal results for the patient in terms of functionality but also from an aesthetic perspective.

## 4. Extended Approaches 

Recent studies report on the extension of the intra-orbital routes to gain enhanced access to the anterior and medial skull base. 

### 4.1. Exposure of the Parasellar and Frontotemporal Regions

Kurbanov and colleagues [55] implemented a three-staged extended version of the classic intra-orbital approach. In stage 1, a superior eyelid TOA approach is described, followed by stage 2 characterized by extradural removal of the lesser sphenoid wing, superior part of the greater sphenoid wing, and anterior orbital roof, with sparing of the posterior orbital roof. Finally, partial osteotomy of the lateral part of the greater sphenoid wing, inferior to the floor of the middle fossa, is achieved in stage 3. This approach allowed for an enhanced exposure of the parasellar and frontotemporal regions including the posterior communicating artery, anterior choroidal artery, and MCA bifurcation. The stage 2 resection provided the biggest increase in exposure of the parasellar region, while the stage 3 resection significantly increased the exposure of the lateral temporal area. 

Overall, this minimally invasive approach improved the visualization of temporal, Sylvian, and parasellar areas compared to the standard transorbital approach, offering significant advantages for surgeries in this area. Moreover, this approach can be coupled to other standard craniotomies such as the pterional craniotomy. 

### 4.2. Extension via a Superolateral Orbital Rim

Lim and colleagues [56] devised an extension of the endoscopic trans-orbital approach via a supero-lateral orbital rim (SLOR) osteotomy. Following a superior endoscopic transorbital approach, including trans-eyelid or below-the-eyebrow incisions, SLOR osteotomy was performed by drilling in the lateral orbital wall, temporal, and frontal bones. Subsequently, anterior clinoidectomy was conducted by drilling through the orbital floor, lesser sphenoid wing, and optic strut. This extended approach was tested on five cadaveric samples and then applied in six clinical cases, comprising three anterior clinoid meningiomas, one chondrosarcoma, and two trigeminal schwannomas. All cases underwent gross total resection (GTR), with no postoperative complications or new-onset neurological deficits. Visual function remained stable in all cases, with one case showing improvement.

In the cadaveric part of the study, the extended approach provided additional visualization of the planum sphenoidale, tuberculum sellae, and the medial part of the contralateral optic nerve, which are not easily accessible through purely endo-orbital or endo-orbital plus lateral orbital ring osteotomy approaches. This enhanced exposure is particularly advantageous for tumors involving the anterior clinoid process and the temporal area. 

Compared to pterional and minipterional craniotomies, this approach is considered less invasive, reducing the risk of temporalis muscle atrophy and ensuring better cosmetic outcomes. Additionally, the extended approach significantly increased vertical and horizontal movements and surgical freedom in the operative field. These advantages are crucial, especially in complex endoscopic procedures involving delicate neurovascular structures in the anterior and middle skull base. While limited by a scarcity of real-life clinical cases and technical constraints of cadaveric anatomical studies, the minimally invasive nature of these extended corridors and their wide access to critical skull base structures warrant further investigation.

## 5. Multiportal Approaches

In recent decades, the EEA has been utilized to address skull base lesions located in the median and paramedian areas [57], along with its extended versions involving complex technical maneuvers [58]. However, these extended approaches carry an increased risk of neurological injuries and postoperative complications due to the challenges in closing created osteodural defects [59,60]. Currently, instead of solely relying on endonasal endoscopy, combining this technique with alternative access routes, such as the transorbital corridor, has been proposed to enhance maneuverability, particularly in the coronal plane. These authors specifically refer to the combination of endoscopic techniques for skull base surgery that we may conventionally call “endoscopic multiportal surgery (EMS)”.

A description of the anatomical area exposed via both pathways is detailed below. Proceeding in a rostro–caudal direction, for the ACF, a combined approach makes it possible to work on both sides of the anterior clinoid process, facilitating the decompression of the optic nerve canal [61]. This proves advantageous for conditions such as spheno-orbital meningiomas [62]. The trans-pterygoid approach, performed through the endonasal route, provides access to the anterior portion of the middle cranial fossa, particularly the space between the first and second branches of the trigeminal nerve [63]. The transorbital route converges in the same region, near Mullan’s triangle [64]. After reaching the anterior and MCF by removing the minor and major wings of the sphenoid, respectively, access to the PCF is limited through the transorbital route due to the sub-temporal ridge and the anteroposterior tilt of the floor of the MCF. However, a corridor within specific anatomical landmarks provides access to the petrous apex, connecting the transorbital and endonasal endoscopic approaches [65]. Drilling the upper portion of the petrous apex, lateral to the mandibular branch of the trigeminal nerve, allows for entry into the posterior cranial fossa through the trigeminal pore. The combination of both approaches allows for up to 97% of the petrous apex to be removed, providing complete access to the PCF [32].

### 5.1. TOA and EA Endoscopic Multiportal Surgery (ETEMS)

Several anatomical studies have demonstrated the feasibility and strengths of the TOA and EA endoscopic multiportal surgery (ETEMS) [66]. Alqahtani et al. [41] described back in 2015 that an ETEMS offers significant value in terms of the extent of exposure and freedom of movement of the hands of the two surgeons, and allows for a better visualization and control of the ventral skull base. Moreover, Guizzardi et al. demonstrated that the areas of connection of the ETEMS were at the level of the sphenoid plane for the ACF; at the level of Mullann’s triangle for the MCF; and finally, just behind the medial portion of the petrous apex for the PCF. The average extradural working areas through the transorbital approach are 4.93, 12.93, and 1.93 cm^2^ and from the endonasal corridor were 7.75, 10.45, and 7.48 cm^2^ at the level of the anterior, middle, and posterior cranial fossae, respectively [67]. In vivo evidence to demonstrate connection areas comes from a single institution study of eight patients treated with ETEMS. The combined approach targeted the cavernous sinus (CS) as the connecting area for four patients with tumors that infiltrated the MCF through the CS. For two patients with MCF tumors that extended into the infratemporal fossa (ITF), the horizontal portion of the greater wing of the sphenoid and the foramen ovale were used as the connection area. In the remaining two patients, the connection was established through the optic canal (OC) [68]. The clinical utility of combined access to middle cranial fossa pathologies was also highlighted by another group that used ETEMS for CS and Meckel’s cave lesions [62]. Not only oncologic pathology but also repair of high-flow CSF leaks from the orbital apex and MCF with a pedicled nasoseptal flap can benefit from an ETEMS [69].

### 5.2. Biportal Transorbital Endoscopic Approach

The biportal transorbital endoscopic approach (BiETOA) to achieve greater surgical freedom has been proposed as a valuable EMS. Lim et al. [70], when comparing the mean maximum angle of attack, observed significant differences between BiETOA and the transorbital endoscopic approach (ETOA) (*p* < 0.01), while no significant differences were found between BiETOA and ETOA lateral orbital rim osteotomy (LOR) (*p* = 0.207, *p* = 0.21). Mean surgical freedom showed significant differences between both BiETOA and ETOA (*p* < 0.01) and LOR BiETOA and ETOA osteotomy (*p* < 0.01). The clinical cases demonstrated successful tumor removal without complications. BiETOA, with its greater surgical freedom and better visibility of deep target lesions, yielded favorable surgical and aesthetic results. Recently, a cadaver study was conducted using the same approach. BiETOA showed limitations in access to bilateral A1 segments and ACoA, with inaccessibility observed in 30% (bTMS) and 60% (bTONES) of exposures. Furthermore, despite the better visualization provided by the transorbital biportal approach, it did not improve surgical freedom [37]. Therefore, overall, although the transorbital biportal approach offers better visualization, its limitations in addressing midline lesions due to the preserved orbital rim restrictions and nevertheless insufficient surgical freedom require further comparative studies to determine the optimal approach that minimizes skull base destruction and maximizes instrument access.

### 5.3. TOA Combined with Extra-Orbital Approaches

Several standard neurosurgical approaches have been described in combination with novel TOA techniques. 

In particular, De Rosa et al. [71] describe the use of a combined endoscopic superior eyelid and extra-orbital approach to access the spheno-orbital region. In their anatomic study on three cadavers, the TOA corridor through the superior eyelid was complemented by an extra-orbital access through the zygomatic bone and the lateral part of the greater sphenoid wing. This resulted in increased surgical freedom, specifically in accessing structures like the SOF, foramen rotundum, and foramen ovale. 

This approach was then used for the surgical management of a 37-year-old female patient with spheno-orbital meningioma, extending from greater and lesser sphenoid wings into the SOF and ipsilateral cavernous sinus. The hybrid technique enabled a 50.63% tumor resection, with no postoperative neurological deficits reported at discharge. 

This study illustrates the technicalities of a dual approach to the anterior skull base and shows promising results in a complex skull base resection case. 

Moreover, Noiphitak et al. [72] demonstrate that combining an endoscopic lateral transorbital route with lateral orbital rim osteotomy in seven cadaveric samples is a feasible solution for enhanced cavernous sinus visualization and offers increased maneuverability in the area. 

In another anatomic study, Matsuo and colleagues [73] report how endoscopic access complements the trans-lateral orbital wall approach, used to access lesions involving the cavernous sinus, and increases the access to the orbital apex and the ipsilateral cavernous sinus, especially around the anterior clinoid process. 

Taken together, these findings show mounting evidence of combined endoscopic transorbital–extra-orbital techniques, showing their feasibility in the case of complex pathologies involving the anterior and middle cranial fossa [74]. Despite cadaveric studies representing a cornerstone of advanced training in these combined techniques and enabling comprehensive learning of skull base anatomy, clinical expertise in real-life cases is paramount. Future efforts should focus on increasing the number of case series to establish specific criteria for approach selection and highlight potential postoperative complications. 

### 5.4. Thetra-Portal Endoscopic Approach

An avant-garde variation on the theme of multiple accesses to the complex regions of the skull base is offered by solutions with three or even four endoscopic ports, configuring interesting parallels with other surgeries such as laparoscopic abdominal surgery.

Trans-pterygoid endonasal, trans-maxillary sublabial, transorbital endoscopic, and transoral endoscopic approaches to access the infratemporal fossa (ITF) were completed on five cadavers in another study. Indeed, the authors showed that a combination of infratemporal approaches with minimal access can provide adequate exposure of the entire ITF while avoiding some of the morbidity associated with open approaches [75]. However, nowadays the applicability of these approaches in clinical practice remains marginal, due to the nonetheless nonzero morbidities and associated technical challenges.

## 6. Limitations and Future Prospectives

The endoscopic transorbital approach represents a frontier in minimally invasive neurosurgery, embodying the ethos of “doing more with less”, and enabling from similar to better surgical goals while mitigating collateral damage. However, there are some limitations. One of the most significant constraints of the endoscopic transorbital route is its steep learning curve, a factor not yet thoroughly studied in the literature [76]. Unlike more conventional surgical approaches, the complexity of this technique demands not only the surgeon’s expertise but also a high level of proficiency from the entire surgical team. The absence of specific studies on the learning curve associated with this approach underscores a gap in current knowledge, suggesting a need for comprehensive training programs and educational frameworks. Furthermore, the physical limitations imposed by the proximity to the ocular globe pose unique challenges. The requirement to minimize pressure on the eye, limiting its displacement to no more than one centimeter, severely restricts maneuverability [3]. This constraint not only complicates the surgical procedure but also increases the risk of complications at the surgical site, including neurovascular damage such as injury to the cavernous sinus or carotid artery, and dural tears. These risks are exacerbated by the narrow surgical corridor, which not only makes it difficult to control complications but also limits the variety of surgical instruments that can be used [77]. 

A future direction that should certainly be pursued is to make the classification of TOAs, and especially their variants, more harmonious according to the point of entry, as it was in Moe’s initial idea, that is, the anatomical region to be reached. In this direction, a recent paper has been published that modularly classifies TOAs conducted through top incision into standard vs. extensions. Extensions of the approach are further divided into those that widen the proximal corridor, which allow for greater maneuverability, and those that widen the distal corridor, which allow for access to various deep anatomical compartments [78]. Certainly, articles such as these represent milestones in improving standardization and making methodological learning easier. 

Despite these challenges, the future of the endoscopic transorbital corridor in minimally invasive neurosurgery is promising. The increasing popularity of endoscopic techniques, both in cranial and spinal surgery, emphasizes a broader shift toward less invasive methods. Further cohort studies are needed to provide valuable data on this approach, refining its indications and potentially expanding its applicability. The evolution toward multi-port endoscopy, integrating biportal endoscopy with endonasal approaches and transorbital approach at the same time, illustrates the potential for hybrid techniques [32]. These innovations promise to expand the operative field and enhance the efficacy of minimally invasive neurosurgery.

## 7. Conclusions

The transorbital approach (TOA) has undeniably revolutionized the treatment of numerous skull base diseases, replacing complex and morbid surgical procedures of the past. However, several considerations must be addressed for its effective implementation. Firstly, we propose a structured training regimen, as outlined in our research, progressing from extradural orbit lesions to deeper intracranial pathologies, including the posterior cranial fossa (PCF). This sequential approach ensures proficiency in increasingly complex procedures. In our study, we provide a comprehensive overview of target pathologies and surgical techniques, ranging from traditional microscope procedures to endoscopic and exoscopic approaches. Nonetheless, it is evident that standardized nomenclature of approaches, agreed-upon anatomical landmarks, shared classifications, and well-designed randomized clinical trials are urgently needed to advance the field. Moreover, we advocate for the integration of multiple minimally invasive endoscopic approaches in clinical practice, particularly for complex pathologies that cannot be adequately addressed by a single approach. This becomes especially pertinent in oncologic neurosurgery, where the versatility of combined approaches can optimize patient outcomes.

## Figures and Tables

**Figure 1 jcm-13-02712-f001:**
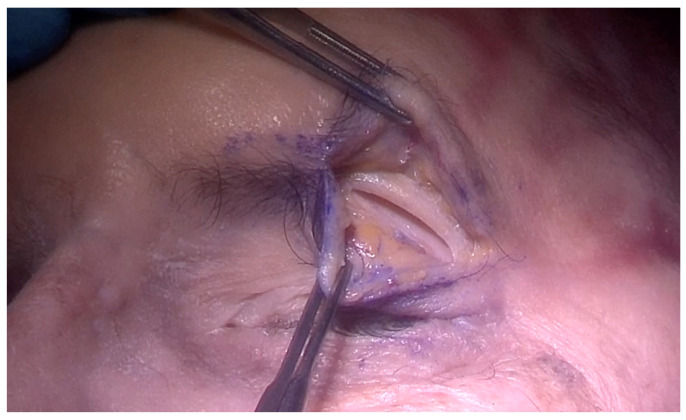
TOA via superior eyelid crease approach; entry point.

## Data Availability

The data presented in this study are available on request from the corresponding author.

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
