# Peer review of "The Transorbital Approach: A Comprehensive Review of Targets, Surgical Techniques, and Multiportal Variants"

_jcm, 2024, doi:10.3390/jcm13092712_

Round 1
Reviewer 1 Report
Comments and Suggestions for Authors
Thank you for inviting me to review this submission titled “The Transorbital Approach: A Comprehensive Review of Targets, Surgical Techniques, and Multiportal Variants”. Here are some comments and suggestions for the authors:
- I agree with the authors that reviewing this topic is interesting and provides paramount information for applying this technique. I’d suggest providing more specific information related to relevant anatomy in the abstract to point to crucial aspects of the topic instead of some information provided in this version. This could enhance and make it more attractive for neurosurgical readers.
- In the same way, the introduction can be focused on the Transorbital Approach (TOA) instead of the EEA. Maybe a historical review of the approach would fit better.
- Double-check all references as it seems that it has 2 numbers in some cases (altered formatting?).
- Go through the entire document to ensure that all abbreviations are used consistently after their first mention. (review TOA-trans orbital approach, Transorbital neuroendoscopic surgery (TONES), etc.).
- The whole manuscript intends to go forward through a classification based on the complexity of cases. I’d suggest introducing briefly the anatomy before starting the rest of the review. Given the anatomical aspects of this topic, a cadaveric dissection or a graphic illustration of the anatomy of the anterior and middle fossas is needed. At least the anatomy of the entry points through the orbit should be illustrated.
- Technical aspects including assessment of each pathology, surgical tenets, and other information are reviewed appropriately.
- Otherwise, this review article encompasses all relevant information regarding all variants of TOAs, including multiportal and micro/exo/endoscopic approaches.
- These minimally invasive techniques are promising and the literature is increasing. Endoscopic surgeons are gaining fields through different angles which may improve surgical outcomes in the coming future.
Comments on the Quality of English Language
Double-check the use of abbreviations all over the manuscript.
Author Response
Thank you for providing your valuable feedback on the manuscript. Here are our responses to each of your points:
-
Abstract Clarity: We have revised the abstract to include more specific information related to relevant anatomy, highlighting crucial aspects of the topic to make it more attractive and informative for neurosurgical readers.
-
Introduction Focus: The introduction has been updated to focus specifically on the Transorbital Approach (TOA), providing a historical review of the TOA to better set the stage for the comprehensive review that follows.
-
Reference Formatting: All references have been carefully reviewed to ensure consistency, and any formatting issues, including instances where there appear to be two numbers, have been addressed.
-
Consistent Abbreviations: We have ensured that all abbreviations, are used consistently throughout the manuscript after their first mention.
-
Anatomical Illustrations:
We put in the introduction the anatomical aspects you requires. A comprehensive anatomical description is not provided because of the impossibility of dealing simplistically with an anatomical approach that actually contains within itself numerous variations. Moreover, it is not our purpose in this review to offer anatomical detail but rather to discuss surgical techniques and offer a kind of scaffold on which to build subsequent studies. The main merit in our opinion of this paper is to offer an overview of what TOA is today
We understand the reviewer's keen interest, which is why we added Figure 1, which shows the entry point through the most frequently used variant of TOA. (Superior eyelid crease approach)
What is more, let us place attention on the whole of section 2 and table 1; indeed, we think that this part is a sufficient anatomical analysis, for the paper's purpose.
-
Comprehensive Coverage: We have confirmed that the technical aspects, including the assessment of each pathology and surgical tenets, are reviewed appropriately. The manuscript continues to encompass all relevant information regarding the variants of TOAs, including multiportal and micro/exo/endoscopic approaches.
-
Future Perspectives: Future advancements and implications of minimally invasive techniques have been emphasized, highlighting the promising nature of these techniques and the increasing literature surrounding them.
Thank you once again for your thoughtful review. We have carefully addressed each of your comments to enhance the quality and clarity of the manuscript.
Reviewer 2 Report
Comments and Suggestions for Authors
Thak you for inviting me to review this manuscript. Authors present a review on a new technique in skull-base neurosurgery. The manuscript is well prepared, and gives a summary to the field. In my opinion, this review will be of great addition to the literature, however, I have some comments:
1- The numbering of sections is incorrect. There is no section 2. Also, section 3 has 3.1, then 3.1.1, 3.1.2, 3.1.3, ... but there is no 3.2. Therefore, this must also be corrected.
2- The last paragraph of the introduction should better explain the objectives and aims of this review. It must be stated in a more straight-forward and a clearer way.
3- This review could benefit from a diagram that summarizes orients the reader about the coming sections.
4- Future prospectives could be more elaborated, with a small highlight on the needed research to improve these techniques.
Finally, I would like to thank the authors on their great work.
Good luck
Comments on the Quality of English Language
Some language flaws are present (especially concerning commas). Needs a final proofread.
Author Response
Thank you for your valuable feedback on our manuscript. We have addressed all the points you raised:
1- We have corrected the numbering of sections and subsections according to your suggestions.
2- The last paragraph of the introduction has been revised to provide a clearer and more concise explanation of the objectives and aims of the review.
3- We have included a diagram to provide an overview and orientation for the reader regarding the upcoming sections.
4- Future prospects have been elaborated further, with a focus on highlighting areas for needed research to enhance these techniques.
We appreciate your acknowledgment of our work and thank you for your time and input.
Round 2
Reviewer 1 Report
Comments and Suggestions for Authors
Thank you for inviting me to review this new version of your manuscript. The authors have taken care of all the suggestions. Again, congrats for your efforts.